# Determination of Quality Parameters in Mangetout (*Pisum sativum* L. ssp. *arvense*) by Using Vis/Near-Infrared Reflectance Spectroscopy

**DOI:** 10.3390/s22114113

**Published:** 2022-05-28

**Authors:** María del Carmen García-García, Emilio Martín-Expósito, Isabel Font, Bárbara del Carmen Martínez-García, Juan A. Fernández, Juan Luis Valenzuela, Pedro Gómez, Mercedes del Río-Celestino

**Affiliations:** 1Department of Agro-Food Engineering and Technology, IFAPA Centro La Mojonera, CAGPDS, 04745 Almería, Spain; emilio.martin.exposito@juntadeandalucia.es; 2ETSIIT, Campus Aynadamar, University of Granada, 18071 Granada, Spain; isabelfont@correo.ugr.es; 3ESI, Higher Engineering School, University of Almería, 04120 Almería, Spain; barbaramgbm@gmail.com; 4Department of Agronomical Engineering, Technical University of Cartagena, 30203 Murcia, Spain; juan.fernandez@upct.es; 5Department of Biology and Geology, Higher Engineering School, University of Almería, 04120 Almería, Spain; jvalenzu@ual.es; 6Department of Plant Breeding and Biotechnology, IFAPA Centro La Mojonera, CAGPDS, 04745 Almería, Spain; pedro.gomez.j@juntadeandalucia.es; 7Agri-Food Laboratory, CAGPDS, Avda, Menéndez Pidal, s/n, 14080 Córdoba, Spain

**Keywords:** mangetout, pea pod, near-infrared reflectance spectroscopy, quality parameters

## Abstract

*Pisum sativum* L. ssp. *arvense*, is colloquially called *tirabeque* or mangetout because it is eaten whole; its pods are recognized as a delicatessen in cooking due to its crunch on the palate and high sweetness. Furthermore, this legume is an important source of protein and antioxidant compounds. Quality control in this species requires the analysis of a large number of samples using costly and laborious conventional methods. For this reason, a non-chemical and rapid technique as near-infrared reflectance spectroscopy (NIRS) was explored to determine its physicochemical quality (color, firmness, total soluble solids, pH, total polyphenols, ascorbic acid and protein content). Pod samples from different cultivars and grown under different fertigation treatments were added to the NIRS analysis to increase spectral and chemical variability in the calibration set. Modified partial least squares regression was used for obtaining the calibration models of these parameters. The coefficients of determination in the external validation ranged from 0.50 to 0.88. The RPD (standard deviation to standard error of prediction ratio) and RER (standard deviation to range) were variable for quality parameters and showed values that were characteristic of equations suitable for quantitative prediction and screening purposes, except for the total soluble solid calibration model.

## 1. Introduction

Vegetable proteins are appearing as a sustainable source for human consumption [1]. Demand for protein is likely to increase significantly over the next few decades to keep pace with a growing population, which is projected to reach nearly ten billion by 2050 [2]. The trend of animal protein consumption is increasing in recent decades [3,4], with production of animal source foods responsible for a significant proportion of global greenhouse gas (GHG) emissions, water consumption and land use [5]. However, the proportion of protein consumption that the World Health Organization recommends is 75% vegetable and 25% animal [6]. In this context, legumes, including soybeans, peanuts, beans, peas, fava beans and lentils, among others, have a higher protein content than most plant foods and about twice the protein content of cereals [7]. The high protein content of legumes may be related to their association with nitrogen-fixing bacteria in their roots, which converts the unusable nitrogen into ammonium that is used for protein synthesis [8]. 

At present, the consumer demands new products on the supermarket shelves and is also attracted by local markets and products. A segment of the population considers itself a green consumer [9], in its different variants, and values healthy and quality foods. Legumes, for all the above exposed, satisfy the current market trends [10].

Several species have been the subject of research for the diversification of vegetables in the agricultural system of the province of Almería (Southeast Spain), with more than 32,000 hectares of greenhouses [11], more than 60% of cultivated vegetables belonging to the *Solanaceae* family. The species tested to diversify these horticultural crops are sweet cucumber, berries, pitahaya, passion fruit, fig tree and a wide range of legumes, among some of them *Pisum sativum* L. ssp. *arvense*, colloquially called *tirabeque* or mangetout [12]. This species is recognized as a delicatessen in cooking due to its crocanti on the palate and high sweetness. Whole mangetout pods are cooked and eaten, this being possible by the absence of “parchment” in the pod walls, hence its pod is indehiscent. The external appearance of pods, particularly their color, is also of great importance when considering the fruits destined for fresh products. 

Previous studies have also revealed the nutritional potential of mangetout, not only for its protein content but also for its content of total soluble solids and antioxidant compounds such as polyphenols, ascorbic acid, fiber, phytoprostanes and phytofurans [12,13,14]. 

Overall, the methodology used for the determination of phenolic compounds and ascorbic acid content is based on spectrophotometric and chromatographic techniques; however, these techniques require expensive equipment and usually use hazardous and pollutant reagents [15,16]. Another relevant method includes colorimetric and titration measurements, since it represents a relatively simple method for measuring total phenolic compounds and ascorbic acid content, respectively. 

The need to carry out screening in breeding programs, quality controls, traceability studies and/or obtaining rapid information for labelling in a large number of samples using conventional methods, leads to high costs, labour input and delays in the rapid decision making. For this reason a non-chemical (producing no chemical waste) and rapid technique, near-infrared reflectance spectroscopy (NIRS), which has been successfully applied in various fields from life sciences to environmental issues, is explored here to screen quality in mangetout pods [17]. Near-infrared spectroscopy is a technique that uses the radiation absorbed by a set of samples in the region from 780 to 2500 nm (near-infrared regionNIR spectroscopy in combination with chemometric analyses can be used for analysis of numerous components (protein, carbohydrates, carotenoid, minerals, glucosinolates, phenolics) and parameters of the sample (firmness, Brix, acidity, color) to be analyzed [18,19,20,21,22,23]. NIRS depends on the number and type of C-H, N-H and O-H bonds in the material being analyzed, then spectral features are combined with reliable compositional or functional analyses of the material in a predictive statistical model. This model is then used to predict the composition of new or unknown samples [24].

Recently, the use of NIRS models for predicting the quality of vegetables has been reported, several of which have addressed zucchini [19,20], pepper, rocket leaves, blackberries [16,21,22] and Ethiopian mustard leaves [23], among others. The seed quality of various legume species has also been analyzed using NIRS such as lentils [25], chickpeas [26] and pea accessions from different germplasm collections [27,28]. Other studies have focused on predicting the sensory quality and maturity of peas [29,30] using NIRS. To the best of our knowledge, there is no research that predicted the quality in mangetout pods.

NIRS calibration models have been developed using a variety of linear regression approaches, including modified partial least squares regression (MPLS). The modified partial least squares (MPLS) is an improved version of traditional PLS that was developed by Shenk and Westerhaus [31]. The MPLS procedure copes more effectively with non-analyte interference in multicomponent determinations. This regression approach is a soft-modeling method for generating predictive models when the factors are many and very collinear. It allows us to develop a model that is then evaluated on external samples to estimate the predictive ability of the model. The mathematical procedure’s end goal is to decrease the large amount of spectral data points (1050 data points from 400 to 2500 nm wavelength range, every 2 nm) and remove the correlation presented by neighboring wavelengths. As a result, the model developed only takes into account the most significant factors, with the “noise” encapsulated in the less important factors, hence the accuracy of NIRS analysis is improved. 

At present, the purpose of the producers and the Andalusian Administrations involved in the cultivation of mangetout is to apply for a “Protected Geographical Indication” (PGI) for the Dalías Valley (Almería, Southeast Spain). This European Indication distinguishes the quality attributes of the products grown in a certain region, and the NIRS technique is a suitable tool that could contribute quickly and accurately to verify the quality of the productions. 

The objective of this paper was to investigate the feasibility for measuring physicochemical quality parameters (color, firmness, total soluble solids, pH, total polyphenols, ascorbic acid and protein content) of mangetout pods by means of VIS-NIRS. For this purpose, different cultivars of mangetout grown under organic cultivation and two fertigation regimes were tested to generate the highest variability for the development of NIRS prediction models.

## 2. Material and Methods

### 2.1. Plant Material

The vegetal material consisted of a local landrace (germplasm maintained by local growers in Almería Province, Southeast Spain) and 7 commercial cultivars of mangetout (Figure 1, Table 1). 

Edible pods of *Pisum sativum* L. spp. *arvense* (tirabeque or mangetout) were grown in an organic greenhouse of 800 m^2^, at Instituto de Investigacion y Formacion Agraria y Pesquera (IFAPA) Center “La Mojonera” (36°48′ N, 2°41′ W; altitude 142 m). The crop (Figure 2) was carried out according to European ecological regulations [32]. The crop cycle took place between October 2020 and March 2021. Two treatments, T100 (100% fertigation treatment) and T50 (50% of water and fertilizers applied), were arranged in a randomized complete block design with 3 replicates, for each cultivar and fertigation treatment, being the planting density of 4 plants per m^2^ [13]. T100 consisted of water and fertilizer provided according to fertigation management. The fertigation treatments allowed us to have a larger number of samples with physicochemical variability (different qualities) to develop NIRS predictive models.

A random monitoring of disease and pest symptoms was conducted weekly. In T100, the consumption of irrigation water was 100 L m^−2^, applying ecological fertilizers so that the average nutrient solution reached 2.3 mS cm^−1^. Pods were harvested when reached standard commercial sizes. 

### 2.2. Physicochemical Parameters

The parameters considered to assess the physical quality in mangetout fruit were firmness and skin color, whereas the parameters of chemical quality were total soluble solids content, pH, total vitamin C, total polyphenol content and protein content. All these characters were determined on the fruit of fresh mangetout except the protein content. For each cultivar (8), treatment (2) and replicate (3), three samples were used. Each sample was composed of 5 pods from 3 plants selected at random, which were then averaged (*n* = 144). 

#### 2.2.1. Firmness

Texturometer XTPlus (Texture Analyzer, Surrey, UK) was used to obtain pod firmness (Figure 3). Shear force was measured by the Warner-Bratzler test. The pod was cut perpendicular with a Warner-Bratzler blade at 1 mm s^−1^ during 5 s. The result was expressed in Newton (N).

#### 2.2.2. Color

CM-700d Konica Minolta portable colorimeter was used. Chroma and Hue angle were measured externally, in two different pod locations, in the central plane. 

#### 2.2.3. Total Soluble Solids and pH

The soluble solid content (TSS) of the pods was obtained through measurement with a Smart-1 digital refractometer (Atago, Japan) (Figure 3), and the previous sample was homogenized for 30 s at 700 Braun CombiMax. The result was expressed in Brix. The pH was obtained by automatic Metrohm 862 Titrosampler (Metrohm, Riverview, Florida, USA) (Figure 3) 

#### 2.2.4. Total Polyphenol Content 

In total, 10 g of the pods was homogenized with 10 mL of ethanol in PT3100 Polytron (Littau, Switzerland) and then centrifuged for 10 min at 4 °C in J2-21M/E Beckman (Fullerton, CA, USA). The pellet was resuspended in 10 mL 70% methanol in water (*v*/*v*) and centrifuged again. Finally, the supernatant was diluted with 25 mL of 70% methanol. This extract was used to determine the TPC according to the Folin–Ciocalteu procedure [33]. In total, 200 μL of the extract, 1 mL of Folin–Ciocalteu solution (diluted 1:10 in water) and 800 μL of Na_2_CO_3_ (7.5%) were mixed vigorously, then the mixture was incubated in the dark at room temperature. After 1 h, absorbance at 765 nm was determined on ThermoSpectronic (Thermo Fisher Scientific, Waltham, MA, USA. The quantification of TPC was expressed in Gallic acid equivalents (mg GAE kg^−1^ Fw).

#### 2.2.5. Vitamin C

The reference values for ascorbic acid content (AAC) were obtained using the iodine titration method by means of an automatic Metrohm 862 Titrosampler [34]. In total, 5 g of sample juice was mixed with distilled deionized water until reaching 50 g of final weight, mixing with 2 mL of glyoxal solution (40%). We proceeded to a brief stirring briefly and 5 min of rest. Once 5 mL of sulfuric acid (25%) was added, it was titrated with iodine (0.01 mol L^−1^) to the end point (EP1). Pure ascorbic acid (AA) was used as an external standard to determine the linearity of the method. For each standard solution, valuations were performed in triplicate. The values of the regression equation and the regression coefficient (r^2^ = 0.9998) were obtained. The ascorbic acid content was expressed as mg 100 g^−1^ fresh weight (fw).

#### 2.2.6. Protein Content

The nitrogen (N) content of the dried and ground pod samples was determined by the Kjeldahl method using a distillation apparatus (k314, Büchi Labortechnik GmbH, Essen, Germany) and then converted to protein content by multiplying it by 6.25. The protein content was expressed as g 100 g^−1^ dry weight (dw). 

### 2.3. Statistical Analysis

Analysis of variance (ANOVA) was used to compare differences among treatments for total marketable yield. Previously, normality and homoscedasticity were tested using the Shapiro–Wilk and Levenne tests, respectively. For these analyses, Fisher’s least significant difference (LSD) test was used to compare the treatments, using the 5% level of significance. Data were analyzed using the Statistical Package for the Social Sciences (SPSS) 24.0 software package (LEAD Technologies, Inc., Chicago, IL, USA).

### 2.4. VIS-NIRS Analysis

Six replicate spectra were recorded for each sample (*n* = 144) and the average of the spectra was calculated. The samples were lyophilized using freeze-drying equipment (Telstar LyoQuest, Terrassa, Spain), then ground in a mill (Janke & Kunkel, model A10, IKA^®^-Labortechnik) for about 20 s to pass through a 0.5 mm screen and stored at −80 °C until analysis. The samples were freeze-dried to eliminate the strong absorbance of water in the infrared spectral region, which overlaps with important bands of nutritional compounds that are present in low concentration. Samples were placed in the NIRS sample holder (3 cm diameter) until it was ¾ full (weight ≅ 3.50 g) and were scanned (Spectrometer Model 6500 Foss-NIRSystems, Inc., Silver Spring, MD, USA). Their NIR spectra were acquired over a wavelength range from 400 to 2500 nm (VIS + NIR regions) at 2 nm intervals.

Principal component analysis (PCA) was used to detect and remove possible spectral outliers (spectra with a standardized Mahalanobis distance (H) from the mean spectrum of the population greater than 3) [35].

Then, laboratory values were added to the spectra files. The reference values were plotted as the dependent variable and the predicted NIRS values plotted as the independent variable. The raw optical data (as log 1/R, being R = reflectance) or first or second derivatives of the log 1/R data, with several combinations of derivative (gap) sizes and segment (smoothing) were used to develop calibration equations [36,37]. Modified partial least squares was used as regression method to correlate the spectral information (raw optical and the different spectral treatments) of the samples and the quality components. The applied pre-treatments to correct baseline offset due to spectral dispersion effects (differences in particle size between samples) were standard normal variate and detrending (SNV-DT) transformations.

### 2.5. Cross-Validation

Cross-validation is an internal validation method [38] and is useful because all samples can be used to perform the calibration equation without the need to maintain separate calibration sets and validation [39]. The method involves dividing the calibration set into M segments (six) and calibrating M times, each time assessing a different part of the set of calibration (1/M) [40]. This number was proposed by WinISI software (Infrasoft International, Port Matilda, PA, USA), five groups being used as the calibration set and then tested on the remaining samples, performing a validation. This process continued until each group of the six was used as a validation group. WinISI software uses principal component analysis as a tool for selecting samples (spectra) to establish the calibration and validation groups. Thus, both groups comprised samples representative of the whole spectral variability of the population with similar mean and standard deviations for each trait.

Thus, cross-validation was conducted on the calibration set to establish the optimum number of terms to be used in building the calibration equations and to identify spectral (H) or chemical (T) outliers. “T” outliers are samples with high residuals when predicted by the model build in the cross-validation. T values of greater than 2.5 are considered significant and those NIR analyses which have large T values may possibly be outliers. The H outlier identifies a sample that is spectrally different from other samples in the population and has a standardized H value of greater than 3.0. The outlier elimination pass was set to allow the software to remove outliers twice before completing the final calibration [41].

The performances of the different calibration equations obtained were determined from cross-validation. Thus, the prediction ability of the equations obtained for each quality component was determined on the basis of two mathematical relationships, which are the standard error of cross-validation (SECV) [42] to standard deviation (SD) ratio (RPD = relative percent difference).

### 2.6. External Validation

To evaluate the precision and accuracy of the equations obtained in the calibration models, an external validation procedure in 30 independent samples was completed. Thus, having ordered the sample set by spectral distance using the CENTER algorithm (Winisi), the 30 samples forming the validation set were selected by taking approximately 1 of every 5 samples in the final 144 sample set. The calibration set thus comprised the remaining 114 samples. 

The statistical methods applied in this study included the coefficient of determination calculated in cross-validation (R^2^ CV) and external validation (R^2^ V), the root mean square error of calibration (RMSEC), the root mean square error of cross-validation (RMSECV) and the root mean square error of prediction (RMSEP). Moreover, the ratio of prediction to deviation (RPD), which indicated the correlations between the SD of the standard wet chemical analyzed data and prediction data by NIRS model (RMSECV or RMSEP) [42], was applied to estimate the prediction ability of the model.

NIR models can be classified depending on the R^2^ value from the external validation [43] as: models (0.26 < R^2^ v < 0.49) with a low correlation; models (0.50 < R^2^ v < 0.64) that can be used to discriminate between low and high values of the samples; models (0.65 < R^2^ v < 0.81) that can be used for rough predictions of samples; models (0.82 < R^2^ v < 0.90) with good correlations; and models (R^2^ v > 0.90) with excellent precision.

The RPD statistic demonstrates how well the calibration model predicts data. The RPD value >3 is desirable for excellent calibration equations, while equations with an RPD <1.5 are unsuitable, according to the guideline used for defining performance calibrations [43]. With regard to the range error ratio (RER), values in the 4 to 8 range indicate the ability to discriminate between high and low values, and RER values from 8 to 12 establish the ability to predict quantitative data [44,45]. 

The mathematical expressions of these statistics are as follows:$$\mathrm{RPD}=SD{\langle {\left[\left(\sum_{i=1}^{n}{\left(y_{i}-{\hat{y}}_{i}\right)}^{2}\right){\left(N-K-1\right)}^{-1}\right]}^{1/2}\rangle }^{-1}$$
where $y_{i}$ = lab reference value for the *i*th sample; $\hat{y}$ = NIR measured value; *N* = number of samples; *K* = number of wavelengths used in an equation; and *SD* = standard deviation.

The coefficient of determination in the cross-validation (R^2^):$$R^{2}=\left(\sum_{i=1}^{n}{\left(\hat{y}-\bar{y}\right)}^{2}\right){\left(\sum_{i=1}^{n}{\left(y_{i}-\bar{y}\right)}^{2}\right)}^{-1}$$
where $\hat{y}$ = NIR measured value; $\bar{y}$= mean “*y*” value for all samples; $y_{i}$ = lab reference value for the *i*th sample.
$$\mathrm{RER}=range{\langle {\left[\left(\sum_{i=1}^{n}{\left(y_{i}-{\hat{y}}_{i}\right)}^{2}\right){\left(N-K-1\right)}^{-1}\right]}^{1/2}\rangle }^{-1}$$
where $y_{i}$ = lab reference value for the ith sample; $\hat{y}$ = NIR measured value; *N* = number of samples; and *K* = number of wavelengths used in an equation.

## 3. Results and Discussion

### 3.1. Marketable Yield

Figure 4 shows the total marketable yield of the diverse varieties in response to different fertigation treatments which ranged from 0.54 to 2.49 kg m^−2^. Significant differences were found between the different cultivars of mangetout, the most productive being the varietal types of the indeterminate climbing growth plant, corresponding to the local Landrace (T50 2.49 kg m^−2^, T100 2.44 kg m^−2^), AR-24009 (T50 2.22 kg m^−2^, T100 2.05 kg m^−2^) and Tirabeque IS (T50 1.76 kg m^−2^, T100 2.05 kg m^−2^), followed by the varieties Tirabí (T50 1.57 kg m^−2^, T100 1.58 kg m^−2^), Pea Zuccola (T50 1.42 kg m^−2^, T100 1.34 kg m^−2^), Capuchino (T50 1.37 kg m^−2^, T100 1.60 kg m^−2^), Pea Delikata (T50 1.37 kg m^−2^, T100 1.34 kg m^−2^) and lastly the Bamby variety which shows the prostrate growth (T50 0.57 kg m^−2^, T100 0.54 kg m^−2^).

The production data obtained in the field trial for most of cultivars were higher than those described previously in mangetout by García-García [46] (0.55–0.65 kg m^−2^), and similar to those indicated by Estrada and Ibáñez [47] (1.5–2 kg m^−2^) in Mediterranean greenhouse conditions.

Increasingly, the use of organic production regulations [48] as well as appropriate fertigation management play an important role in enhancing crop quality and economizing water [13] according to the Sustainable Development Goals (SDGs) by 2030. In this regard, previous studies have showed that yield and quality of snap pods can be significantly affected by different compositions of fertilization [49,50] and by different doses of water in the fertigation solution [13,51]. In order to obtain the highest possible physicochemical variability to develop NIRS predictive models, two fertigation treatments and different mangetout cultivars were used. 

### 3.2. Physicochemical Profiles

The samples analyzed varied in all variables as shown by the range and coefficient of variation (CV) of the calibration set (Table 2). The highest values for the CV were observed for C* chromatic value, firmness, ascorbic acid content and total polyphenol content (>20%), possibly due to the different fertigation treatments and varieties used. 

Based on the results of this study, the chromatic parameters (C* and h*) varied from 15.20 to 35.58 and 105.13 to 112.91, respectively. The h* values correspond to the color green. Green color of fresh pods is one the key factors for deciding the commercial acceptance of snap bean as a fresh vegetable. Similar results have been previously found in snap pods with values ranged from 107 to 111 for h* parameter, but a narrow variation range (27 to 33) for C* chromatic parameter [13,52,53]. 

Texture is a quality attribute in mangetout fruits very important for consumers since its singular quality of edible crunchy pod is highly appreciated. From our study, the firmness values in mangetout pods ranged from 20.59 to 67.52 N. Although information is lacking for the comparison of firmness with other mangetout cultivars from the literature, our previous research results showed that mangetout “Tirabí” showed values included in the range mentioned above [13].

Vitamin C is essential in both plants and animals. The main suppliers of this vitamin in the diet are fruits and vegetables [54]. Legumes are considered an important source of vitamins, especially rich in ascorbic acid content in the pods [55]. Considerable variation was found for AAC which ranged from 19.75 to 68.86 mg AA 100 g^−1^ fw in mangetout pods. Previous studies revealed AA content within the range of AAC showed in this work for three pea varieties (26 to 31 mg AA 100 g^−1^) [56]. Our findings are also in agreement with those of Rickman et al. [57] and Avilés and Cruz [58], who described AA values of 40 and 27 mg 100 g^−1^ fw in peas and pea pods, respectively. Mangetout pods can be considered a rich source of vitamin C, since orange and lemon contain 30–50 mg of ascorbic acid 100 g^−1^ fw [54].

The pH of foods is an important parameter related to the taste perceived by consumers. In our study, the pH ranged from 5.99 to 8.85. The values obtained agree with previous studies on legumes [59,60,61], but lower than those obtained in French bean pods (5.84–5.96) by Segura et al. [62].

TSS is another taste quality determinant [63], and cultivars with higher TSS have higher taste quality. Mangetout pods are rich in TSS content (6.29–8.83 Brix) in comparison with other legume pods; thus, cowpea accessions from different Mediterranean countries showed lower sweetness (range 5.07–7.57 Brix) [55] in relation to our results. 

On the other hand, the fresh mangetout pod TSS content in our study was lower compared to those previously reported in the scientific literature [12]. This previous work revealed that the TSS of fresh pods ranged from 9.1 to 11.3 under specific fertigation treatments demonstrating that the environmental factors such as available water had a highly significant effect on this quality parameter. According to the Brix reference values of the main greenhouse vegetables, the mangetout pods analyzed showed a higher sweetness than California green pepper fruits (4.03–6.31 Brix) and similar to red California pepper fruits (7.37–8.85 Brix) [64]. 

The presence of polyphenols in plants is very varied, depending on the plant species, variety, part of the plant, growing conditions, etc. More than 8000 phenolic compounds with a very varied structure have been identified from simple molecules, such as phenolic acids and complex polymers of high molecular mass such as tannins [65]. In our study, mangetout exhibited higher total polyphenol content (202.30 to 685.05 mg GAE kg^−1^ fw) that those reported for other snap pods, such as the French bean with 300 mg GAE kg^−1^ fw [46]. Our results agree with those of Devi et al. [66] who found a wide variation range (126.3–1286.3 mg GAE kg^−1^ fw) in pea pods from 22 different genotypes. On the other hand, the consumer increasingly appreciates fruits with antioxidant properties due to the health benefits. A source of phenolic compounds is identified as a chemopreventive agent since it eliminates free radicals and has a preventive effect on degenerative diseases, among others [67]. Mangetout pods have a high potential to be used in the development of functional foods or nutraceutical products and unlike pea pods they would not require any processing as the whole pod is edible. 

Our results showed a wide variability for protein content (11.50–29.75 g 100 g^−1^ dw) and our results agree with those of Hood-Niefer et al. [68] (24.4 to 27.5 g 100 g^−1^ dw), but are higher than the results obtained by Mateos-Aparicio et al. [69] (10.8 ± 0.3 g 100 g^−1^ dw) in pea pods. Overall, in the pea, both the seeds (20.5–22.6%) and pods (13.37%) are a rich source of protein [70]. A diet rich in vegetable protein is increasingly important nowadays due to its health benefits and thus it is recommended that people reduce their consumption of animal protein. In addition, pea pods have protein-denaturing properties that show anti-inflammatory effects and anti-cholinesterase activity because of the strong antidiabetic properties of peas [71]. 

### 3.3. VIS-NIRS Analysis

#### 3.3.1. Raw Spectra on Mangetout

Raw spectra of the calibration set samples are shown in Figure 5. A remarkable variability in the VIS region (400–850 nm) absorbance spectra was observed because of pigments. The peak around 640–700 nm illustrated the color transition of pea pods correlated with the chlorophyll content that absorbs radiation in this region [72].

Absorption bands in the region from 1300 to 2000 nm have been assigned to the third overtones of C-N (amines); C=O (ketones, amino acids); and C-O (long-chain fatty acids, phenols). From 2200 to 2400 nm, absorptions of C-N (primary amines) and C-O (alcohols) have been assigned to the third overtones of these compounds, while in the same region, C-H (asymmetrical deformation) and C-O (symmetrical vibrations) have been assigned to the second overtones of these molecules. Finally, the second overtones of C-H deformation and C-N (amides) have been reported in the 2400–2500 nm region [73].

#### 3.3.2. Second Derivative Spectra of Mangetout 

The second derivative and SNT-DT (standard normal variate and de-trending) algorithms to the raw spectra led to a substantial correction (Figure 6) of the baseline shift produced by differences in path length and particle size. The increase in the complexity of the derivative spectra resulted in a clear separation between peaks which overlap in the raw spectra. 

Absorption maxima bands (λmax) were observed between 400 and 700 nm (at 444, 546 nm and 670 nm) in the spectra attributed to pod pigments that absorb in visible region (Figure 6). From all pigments that can be found in plants, chlorophylls are used for photosynthesis (“a” and “b”), which absorb preferentially violet-blue light (400–500 nm) and red light (600–700 nm), respectively [74]. 

Pigment–protein complex molecules could be responsible for some of the traits that determine the VIS region at longer wavelengths. Thus, binding proteins in chlorophyll *a*/*b* absorb in the 498–568 nm range [74] and red absorbing pigments, particularly chlorophyll, give the fruit its green color [75,76]. 

In the region NIRS of the spectra, peaks at 1208 nm (attributed to a CH second overtone), 1726 nm and 1762 nm (assigned to CH first overtone), 2308 nm and 2348 nm (attributed to CH stretch and deformation in a CH_2_ group) were detected which are related to lipids [77,78]. Other peaks located at 1210 nm corresponded to absorption by OH groups in carbohydrates [79,80]. Other peaks at 1512, 2056 and 2174 nm related to protein, specifically to NH stretch, NH stretch and amide II, and amide I and amide III, respectively [78]. The last significant peaks were observed at 1436 and 2270 nm, these wavelengths corresponding to the deformation of the OH + CO cellulose groups [79]. 

#### 3.3.3. Calibration Development 

Table 3 and Table 4 show the summary of the statistics obtained from calibration, cross-validation and external validation models in mangetout samples, respectively. The full available visible region and near-infrared region (400–2500 nm) were used. 

The coefficients of determination (R^2^) achieved in calibration were higher than those found in external validation models for mangetout, as expected. The coefficient of determination for cross-validation (R^2^ CV), oscillated between 0.55 for pH to 0.92 for protein (Table 3), whereas RPDcv values ranges from 1.50 for pH to 3.45 for protein.

Based on the R^2^ values of the external validation, the models were as follows [39]: models that can be used to discriminate between low and high values of the samples (0.50 m< R^2^ < 0.64), in our work the models developed for AAC and TSS; models that can be used for rough predictions of samples (0.65 < R^2^ < 0.81), in our case the calibrations achieved for C* and h* color parameters, firmness and pH; and models with good correlations (0.83 < R^2^ < 0.90), these values corresponding with models obtained for total polyphenol content and protein. 

The SEP values of the validation were lower than their respective SD, which indicates that NIRS is able to determine these traits in mangetout. 

According to the guideline used for defining performance calibrations [43] when this ratio is greater than 3, the calibration equation is very significant, and this was reached in our study for protein content; if RPD values range between 2.5 < RPD < 3, predictive models are considered very good, in our case corresponding to the TPC model; while RPD range between 1.5 < RPD < 2.5 predictive models are appropriate for screening purposes, which was achieved for AAC, pH, firmness and C* and h* color parameter models.

Figure 7 shows the relationship between the predicted reflectance spectroscopy in the near infrared (NIRS) and reference values for all parameters (color parameters (chroma * and hue angle), firmness, total soluble solids, pH, protein content, ascorbic acid and total polyphenol content) in the mangetout validation set samples.

In reference to RER (ratio of the range to standard error of prediction) coefficients, this dimensionless parameter is also used to evaluate the predictive ability of NIRS equations, in this work ranged from 4.84–14.89.

Prediction models for C*, h*, firmness, TSS, pH, AAC showed RER values within the range from 4 to 8, which suggest the possibility of discriminating between high and low values; while RER values in the range of 8 to 12 represent the possibility of predicting quantitative data [44,45] which was achieved for protein and TPC predictive models.

Previous works have demonstrated the validity of the NIRS technique in evaluating the accuracy of pea single seed protein with R^2^ = 0.94 and RPD = 3.7 in external validation [28], and also for predicting soybean single seed protein content with Rval^2^ = 0.84 and RPDval = 2.28 values [81]. 

The estimation of protein and total polyphenol content in common beans (*Phaseolus vulgaris* L.) by NIRS has also been previously assayed by several authors reaching significantly good results in general. Thus, the high R^2^ obtained ranged from 0.91–0.94 and RPD values above 3.5 [82,83,84,85]. Other authors supported the validity of the NIRS technique in similar approaches, with R^2^ and RPD values for firmness of 0.61 and 1.7, respectively, in soybean single seed [82]. Wang [86] used NIRS to predict the total polyphenol content in ground faba bean (*Vicia faba* L.), with an R^2^ of 0.79, RMSECV of 0.40 and RPD of 2.20, and also for the determination of protein in ground faba bean seed powder with an R^2^ of 0.94. 

It should be noted that the prediction accuracies in all of the above-mentioned studies were comparable to those reported for mangetout in this study. To our knowledge, this is the first article dealing with the use of NIRS to predict pod quality traits in mangetout. 

Modified PLS regression was employed to reduce the spectral information of the mangetout samples by creating a much smaller number of new orthogonal variables (factors) which retain the essential information needed to predict the composition of the samples.

#### 3.3.4. Modified Partial Least Squares Loadings for Quality Equations

The scores of the best models for all quality parameters were plotted by their first MPLS loadings (Figure 8) to identify those areas within the spectral range where variance had influenced the model fitting, to a lesser or greater degree, as well as the direction (negative or positive). 

The region of the spectrum which most influenced the fitting of the model was the visible segment between 480 to 700 nm. Thus, the contribution of chlorophyll (672 nm) showed the highest weight on first MPLS loading [75] (Figure 8). Other chromophores absorbing at 496 and 512 nm also participated in the equations. With respect to NIR region, previous studies have shown the contribution of this region to predict color parameters for species such as fresh *Ginkgo biloba* leaves [87], green-leafy species [88] or *Sassafras tzumu* [89]. Some plant chemical compounds (e.g., phenolics and flavonoids) respond to the stress and environmental changes and correlate in a secondary way with the color parameters. The characteristic bands of phenolics and flavonoids can be detected in wavelength regions from 1415 nm to 1512 nm, 1650 to 1750 nm and from 1955 to 2035 nm in the MPLS loadings for the color parameters (Figure 8) [90]. Furthermore, the color is caused by the reflection of helicoidally stacked cellulose microfibrils that form multilayers in the cell walls of the epicarp [91]. Thus, the wavelengths at 1932 nm (O-H stretching plus O-H deformation) could be related to the cellulose of the pod tissues which can be observed in the MPLS loadings of the optimal calibrations for the color parameters (Figure 8). Others main NIR contributions were those at 2284 nm (C-H stretching plus C-H deformation), 2300 nm (stretching–bending of CH—CH2 bonds and C—O bonds) and 2348 nm (C-H combination of methylene groups) [77]. In addition, absorption bands in the NIR region that influenced the fitting of the models were found at 1212, 1388, 1412 and 1990 nm (associated with glucides and water absorptions), and the region around 2072 nm (N—H bonds) associated with protein.

## 4. Conclusions

This work has showed that genetic variability exists for the quality parameters analyzed in mangetout cultivars. Many of the traits analyzed are of economic interest (color, firmness, protein content and antioxidant compounds). These new understandings could be useful in selecting parents for breeding programs aimed at enhancing physicochemical parameters that respond to the new trends market. 

Moreover, the result of the present investigation explores the potential of NIRS to simultaneously determine eight quality traits in mangetout, as an alternative to reference methods. The measurements with the reference methods of most of these parameters are expensive, have laborious protocols and require a long analysis time. Utilizing NIRS, every 2 min, we can analyze all the quality parameters of a sample. The results reveal that the models allow an accurate quantification of protein and TPC and a rough screening method of the samples for color parameters (c* and h*), firmness, AAC and pH. 

The inclusion future of mangetout cultivars from different geographical origins and segregant populations in the calibration models will allow us to increase the robustness of the equations for these parameters.

The performance of the calibration model for TSS was lower than that obtained for the other quality parameters in this work. The low variability among mangetout cultivars used in this work (6.29–8.76 Brix) could be based on the lower accuracy of the calibration model for TSS. An increase in both the number of samples and trait variation can be crucial factors for improving the accuracy of this calibration model. 

It is interesting to focus attention on firmness (shear force). Pod firmness is an excellent indicator of pod quality, but its quantification is time consuming and not easily measured. Pods must be harvested before they become tough and develop poor culinary acceptance, even if it means sacrificing maximum yield. From this point of view, the use of NIRS instead of a texturometer could be clearly advantageous.

Spectral ranges associated with the absorbance of chromophores, carbohydrates, water and protein were used by MPLS regression for the model fitting of quality equations in mangetout.

## Figures and Tables

**Figure 1 sensors-22-04113-f001:**
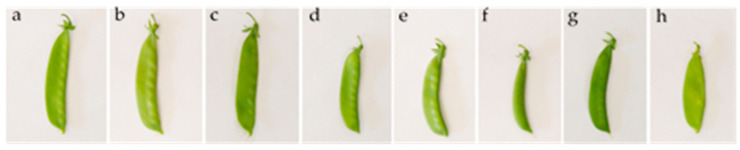
Pea pods of the different cultivars of mangetout analyzed. From left to right: Local landrace (**a**), AR-24007 (**b**), Capuchino (**c**), Tirabeque IS (**d**), Tirabí (**e**), Pea Zuccola (**f**), Pea Delikata (**g**) and Bamby (**h**).

**Figure 2 sensors-22-04113-f002:**
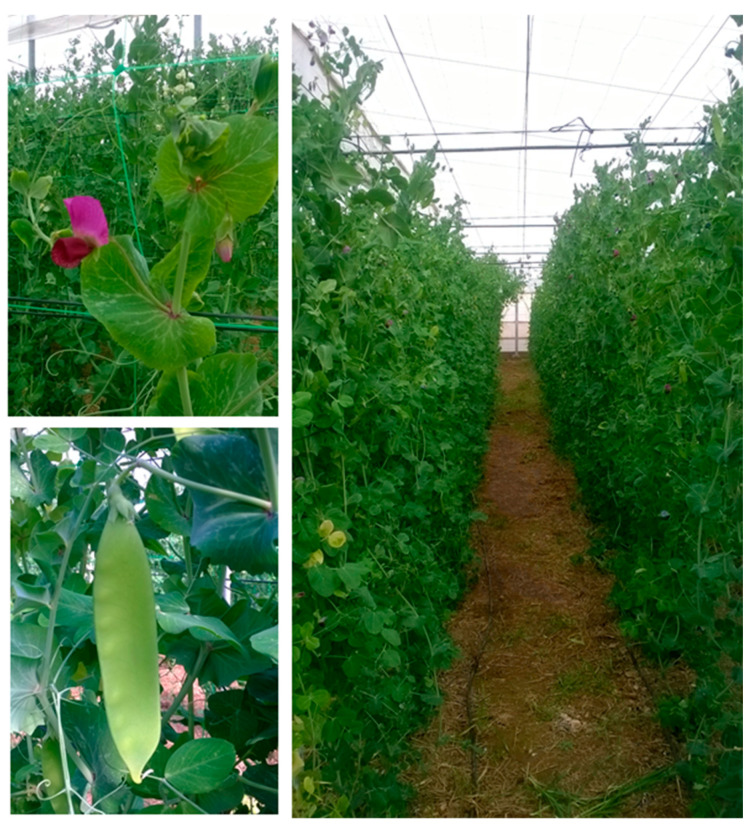
Detail of flower, leaves (**left up**) and pod (**left down**) of mangetout. Panoramic view of field trial (**right**).

**Figure 3 sensors-22-04113-f003:**
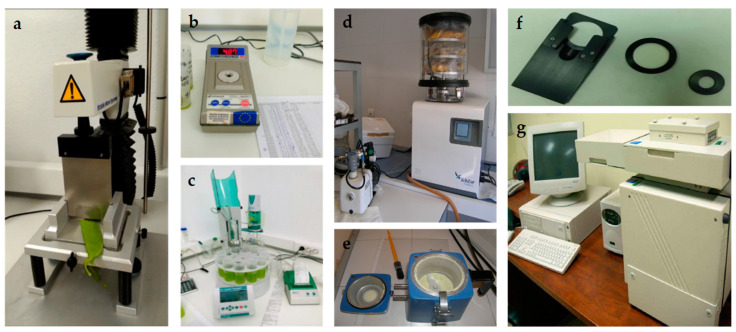
Detail of Texturometer XTPlus Texture Analyzer (**a**); Smart-1 digital refractometer (**b**); Automatic Metrohm 862 Titrosampler (**c**); Freeze-drying equipment (**d**); Mill (**e**); NIRS sample holder (**f**); Spectrometer Model 6500 Foss-NIRSystems (**g**).

**Figure 4 sensors-22-04113-f004:**
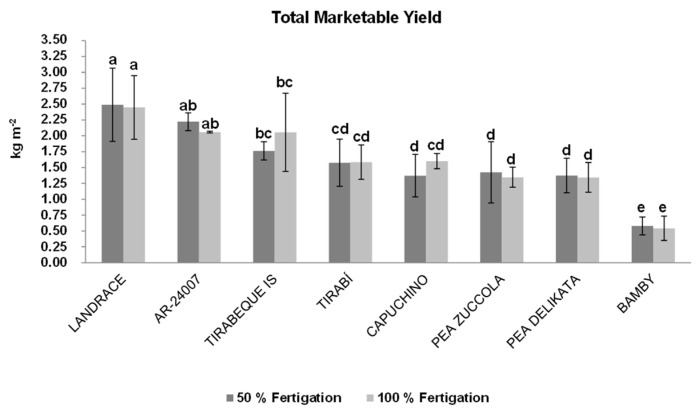
Total marketable yield (kg m^−2^) of the different varieties of mangetout under different fertigation treatments (T50 and T100). Bars with different lowercase letters were significantly different at *p* < 0.05 (Tukey’s multiple range test).

**Figure 5 sensors-22-04113-f005:**
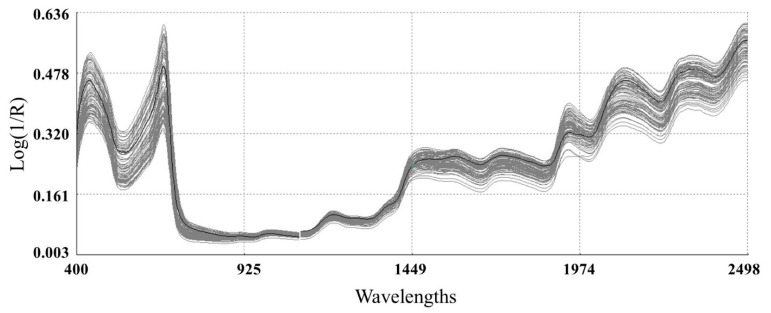
Raw spectra for dried mangetout samples.

**Figure 6 sensors-22-04113-f006:**
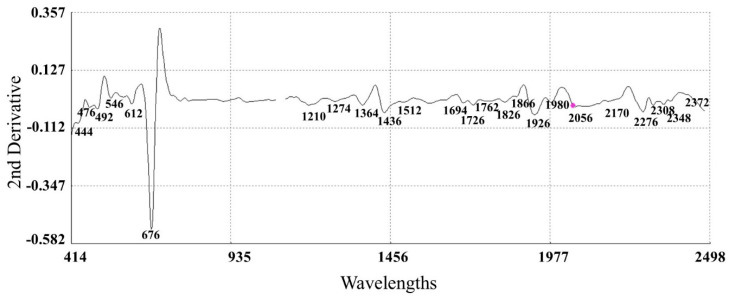
Second derivative spectra (2, 5, 5, 2; SNV + DT) of the raw optical data for mangetout samples in the range of 400 to 2500 nm, together with the most relevant absorption bands.

**Figure 7 sensors-22-04113-f007:**
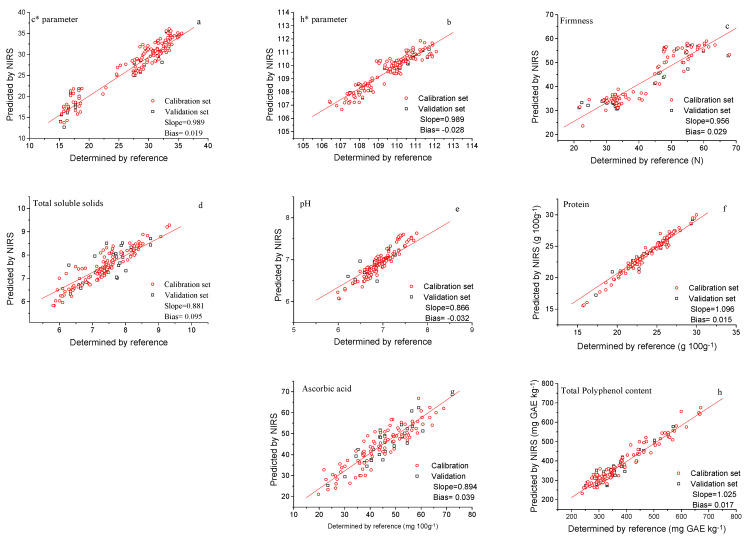
Predicted versus reference values for calibration and external validation for all parameters: c* (**a**); h* (**b**); firmness (**c**); total soluble solids (**d**); pH (**e**); protein content (**f**); ascorbic acid (**g**); total polyphenol content in the mangetout (**h**).

**Figure 8 sensors-22-04113-f008:**
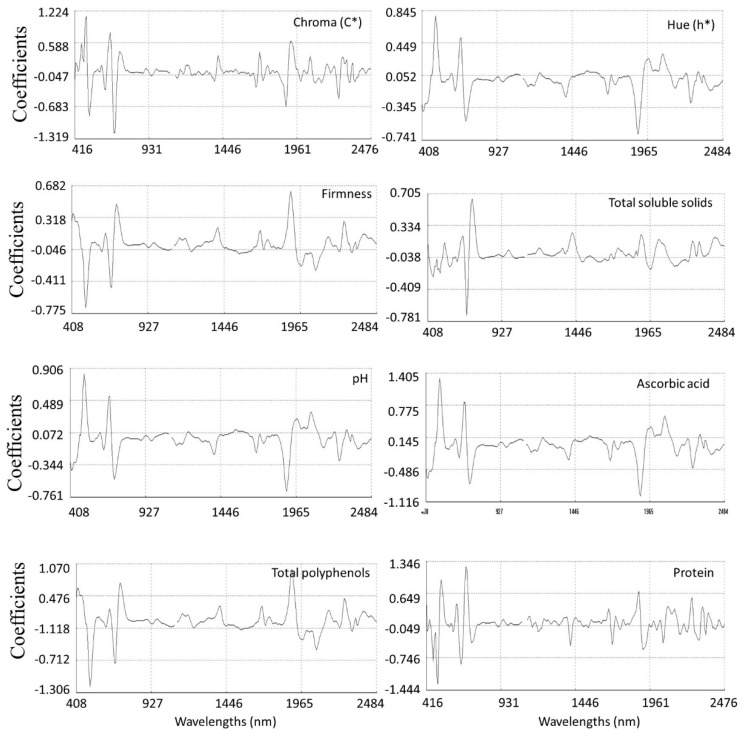
Modified partial least squares (MPLS) loading of the optimal calibrations for physicochemical compounds measured by NIRS.

**Table 1 sensors-22-04113-t001:** Cultivars, companies and growth habit of mangetout used in this study.

Cultivars	Companies	Growth Habit
Local landrace	Growers production	Indeterminate climbing
AR-24007	Ramiro Arnedo	Indeterminate climbing
Capuchino	Batlle	Indeterminate climbing
Tirabeque IS	Intersemillas	Indeterminate climbing
Tirabí	Fitó	Indeterminate climbing
Pea Zuccola	Tozer	Determinate climbing
Pea Delikata	Tozer	Determinate climbing
Bamby	Gautier	Deterninate postrate

**Table 2 sensors-22-04113-t002:** Mean, range and standard deviation (*n* = 144) for quality parameters of the mangetout samples used in this study.

Parameters	Mean	Range	SD
C*	27.87	15.20–35.58	6.15
h*	109.46	105.13–112.91	1.49
Firmness (N)	43.62	20.59–67.52	12.74
TSS (Brix)	7.53	6.08–8.85	0.65
pH	6.80	5.99–7.28	0.27
Protein (g 100 g^−1^ dw)	23.48	11.50–29.75	3.02
AAC (mg 100 g^−1^ fw)	43.82	19.75–68.86	10.82
TPC (mg GAE kg^−1^ fw)	389.09	202.30–685.05	111.52

**Table 3 sensors-22-04113-t003:** Calibration and cross-validation statistics of quality compounds for mangetout.

Parameters	Range	^1^ SD	^2^ R^2^	^3^ SEC	^4^ R^2^ CV	^5^ SECV	^6^ RPDcv	^7^ Treatment	^8^ Cv
C*	15.20–35.58	6.35	0.87	2.24	0.81	2.78	2.28	2,5,5,2	0.22
h*	106.41–112.10	1.41	0.80	0.62	0.71	0.75	1.88	1,4,4,1	0.01
Firmness (N)	21.75–67.52	10.09	0.71	5.46	0.71	5.93	1.70	1,4,4,1	0.21
^9^ TSS (Brix)	6.29–8.83	0.65	0.93	0.18	0.68	0.39	1.66	1,4,4,1	0.08
pH	6.01–7.28	0.27	0.60	0.17	0.55	0.18	1.50	1,4,4,1	0.04
Protein (g 100 g^−1^ dw)	15.69–29.75	2.80	0.97	0.48	0.92	0.81	3.45	2,5,5,2	0.13
^10^ AAC (mg 100 g^−1^ fw)	19.75–64.40	10.89	0.79	5.02	0.56	7.16	1.52	1,4,4,1	0.24
^11^ TPC (mg GAE kg^−1^ fw)	239.28–670.30	101.91	0.93	27.01	0.86	39.08	2.61	1,4,4,1	0.28

^1^ SD: Standard deviation; ^2^ R^2^: Coefficient of determination in calibration; ^3^ SEC: Standard error in calibration; ^4^ R^2^: Coefficient of determination in cross-validation; ^5^ SECV: Standard error of cross-validation; ^6^ RPDcv: Ratio of the standard deviation to standard error of cross-validation; ^7^ Mathematical treatment; ^8^ Coefficient of variation; ^9^ TSS: Total soluble solids; ^10^ AAC: Ascorbic acid content; ^11^ TPC: Total polyphenol content.

**Table 4 sensors-22-04113-t004:** Reference values and external validation statistics of the NIRS calibrations for quality compounds in mangetout.

	Reference Values (*n* = 30)				External Validation		
Parameters	Range	Mean	^1^ SD	^2^ Rv2	^3^ SEP	^4^ RPDp	^5^ RER
C*	15.20–34.89	25.50	7.33	0.78	3.34	2.19	5.89
H*	107.40–111.71	109.71	1.24	0.68	0.56	2.00	6.95
Firmness (N)	24.45–67.20	40.48	12.51	0.65	7.34	1.70	5.96
^6^ TSS (Brix)	6.29–8.76	7.54	0.69	0.52	0.51	1.35	4.84
pH	6.22–7.20	6.83	0.22	0.50	0.14	1.57	7.00
Protein (g 100 g^−1^ dw)	17.22–29.5	24.95	2.18	0.88	0.68	3.20	14.89
^7^AAC (mg 100 g^−1^ fw)	22.71–63.47	45.69	8.82	0.50	8.82	1.50	7.03
^8^ TPC (mg GAE kg^−1^ fw)	250.89–570.21	360.89	80.37	0.84	29.46	2.72	10.84

TSS: Total soluble solids; AAC: Ascorbic acid content; TPC: Total polyphenol content; **^1^** SD: Standard deviation; **^2^** Rv^2^: Coefficient of determination in external validation; **^3^** SEP: Standard error of prediction corrected for bias; **^4^** RPDp: Ratio of the standard deviation to standard error of prediction (performance); **^5^** RER: Ratio of the range to standard error of prediction (performance); ^6^ TSS: Total soluble solids; ^7^ AAC: Ascorbic acid content; ^8^ TPC: Total polyphenol content.

## Data Availability

Data sharing not applicable.

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
