# Peer review of "Determination of Quality Parameters in Mangetout (Pisum sativum L. ssp. arvense) by Using Vis/Near-Infrared Reflectance Spectroscopy"

_sensors, 2022, doi:10.3390/s22114113_

Round 1

Reviewer 1 Report

The paper deals with an important and interesting topic of the prediction of the physicochemical composition of magnetout using VIS-NIR. However, the paper includes numerous issues and questions to answer, therefor in this way it is not applicable for publication. 

The introduction is quite good and detailed. Some part (noted in the paper attached) does not fit into it. I suggest deleting it from there (118-120). 
In some cases, I miss the citations also (line 88, line 96). 

The materials and methods are also dealing with some issues. However, my biggest problem is that the authors state that all the measurements were performed on the fresh samples, but the NIR measurements were done on the dried samples (at what temperature how long?). I am quite sure the drying process changes the composition of the sample (Vitamin C, total polyphenol TSS, and so on), which reflects in the NIR spectra also. Therefore this way it is not acceptable. Did the authors check how much it changes after the drying? It does not have the sense to predict in the fresh if it was measured with NIR in the dried sample. 

I also miss some information on the performance of the ANOVA analysis. Find more details in the paper attached. Did the authors test the normality? ANOVA can be only computed if the normality is good. Moreover, for choosing the optimal post-hoc test homogeneity of the variances (Levene-test) must be computed. Tukey test can be used only if it is correct (based on mean >0.05). It can be easily done using SPSS. If homogeneity is not assumed, the Games-Howell test could be used! Moreover, authors should consider using MANOVA instead of ANOVA as we speak about composition. For what group variable was the ANOVA performed, please state it! Moreover, for me, it is not clear how the n=144 came out. 

I also don't understand the presentation of the results. In my opinion, a table would be needed to show the parameters (average, SD) and then another set of tables for the results of the PLSR model. Moreover, in figure 3 there is no SD stated... 
Table 2 should be replaced with the section of NIR separated the way written above!
The quality of figure 7 is very poor!

After the corrections and detailed explanation the paper may be accepted.

Author Response

REVIEWER 1

The paper deals with an important and interesting topic of the prediction of the physicochemical composition of mangetout using VIS-NIR. However, the paper includes numerous issues and questions to answer, therefor in this way it is not applicable for publication. 

We really appreciate the comments and suggestion of the reviewer providing valuable feedback on our manuscript.

For the provided specific points we have made the requested modifications and hope the revised version would meet the requirements.

The introduction is quite good and detailed. Some part (noted in the paper attached) does not fit into it. I suggest deleting it from there (118-120). 

            That is done

In some cases, I miss the citations also (line 88, line 96). 

The authors have added the citations as follow:

Line 88. Burns, D.A., Ciurczak, E.W. Handbook of Near-Infrared Analysis, 3rd ed.; Dekker Inc.: New York, NY, USA, 2008.

Line 96. Murray, I., Williams, D. Chemical principals of near-infrared technology. In Near Infrared Technology in the Agricultural and Food Industries. American Association of Cereal Chemists, St Paul, MN, 1987, 143–167.

The materials and methods are also dealing with some issues. However, my biggest problem is that the authors state that all the measurements were performed on the fresh samples, but the NIR measurements were done on the dried samples (at what temperature how long?). I am quite sure the drying process changes the composition of the sample (Vitamin C, total polyphenol TSS, and so on), which reflects in the NIR spectra also. Therefore this way it is not acceptable. Did the authors check how much it changes after the drying? It does not have the sense to predict in the fresh if it was measured with NIR in the dried sample. 

The authors have included the information about the dried samples in the revised manuscript.

The samples were lyophilized using freeze-drying equipment (Telstar LyoQuest, Germany), then ground in a mill (Janke & Kunkel, model A10, IKA®-Labortechnik) for about 20 s to pass through a 0.5mm screen, and stored at −80 ∘C until analysis.   The samples were freeze-dried to eliminate the strong absorbance of water in the infrared spectral region, which overlaps with important bands of nutritional compounds that are present in low concentration (lines 261-266).

The shorter exposure of nutrients to a minimal presence of oxygen during optimum freeze-drying conditions is less favourable to oxidation/degradation reactions and contributes to the preservation of nutrients and bioactive compounds. We have checked and used this freeze-drying procedure here and other food matrices as zucchini, pepper and blackberries :

-Blanco-Díaz MT, Del Río-Celestino M, Martínez-Valdivieso D, Font R. Use of visible and near-infrared spectroscopy for predicting antioxidant compounds in summer squash (Cucurbita pepo ssp pepo). Food Chem. 2014 Dec 1;164:301-308

-Martínez-Valdivieso D., Font R., Blanco-Díaz M.T., Moreno-Rojas J.M., Gómez P., Alonso-Moraga A., Del Río-Celestino M. (2014). Application of near-infrared reflectance spectroscopy for predicting carotenoid content in summer squash fruit. Computers and Electronics in Agriculture, 108: 71-79

-Eva María Toledo-Martín, María Carmen García-García, Rafael Font, José Manuel Moreno-Rojas, Pedro Gómez, María Salinas-Navarro, and Mercedes Del Río-Celestino. Application of visible/near-infrared reflectance spectroscopy for predicting internal and external quality in pepper”, Journal of the Science of Food and Agriculture, 96(9). 3114-3125. 2015.

-Venyaminov SY and Prendergast FG, Water (H2O and D2O) molar absorptivity in the 1000–4000 cm−1 range and quantitative infrared spectroscopy of aqueous solutions. Anal Biochem 248:234–245 (1997).

I also miss some information on the performance of the ANOVA analysis. Find more details in the paper attached. Did the authors test the normality? ANOVA can be only computed if the normality is good. Moreover, for choosing the optimal post-hoc test homogeneity of the variances (Levene-test) must be computed. Tukey test can be used only if it is correct (based on mean >0.05). It can be easily done using SPSS. If homogeneity is not assumed, the Games-Howell test could be used! Moreover, authors should consider using MANOVA instead of ANOVA as we speak about composition. For what group variable was the ANOVA performed, please state it! Moreover, for me, it is not clear how the n=144 came out. 

The authors has included in the manuscript the following paragraph:

Analysis of variance ANOVA was used to compare differences among treatments for total marketable yield. Previously, normality and homocedasticity were tested using the Shapiro-Wilk and Levenne tests, respectively. For these analyses, Fisher´s least significant difference (LSD) test was used to make comparisons of treatments of most interest, using the 5% level of significance (lines 251-258)

I also don't understand the presentation of the results. In my opinion, a table would be needed to show the parameters (average, SD) and then another set of tables for the results of the PLSR model.

 According to the reviewer's suggestions a table with the means and SD for all the parameters was included in the manuscript (Table 2, line 402)

Moreover, in figure 3 there is no SD stated... 

That is done. Figure 3 has been renamed Figure 4 in the revised version. (line 375)

Table 2 should be replaced with the section of NIR separated the way written above!

That was done. Table 2 has been renamed Table 3 in the revised version. (line 528)

The quality of figure 7 is very poor!

The quality of Figure has been improved. Figure 7 has been renamed Figure 8 in the revised version (line 626).

After the corrections and detailed explanation the paper may be accepted.

We provide a point-by-point response to the reviewer’s comments of the pdf file.

The requirements of the reviewer expressed in the pdf file throughout the article have been answered; some of them are detailed below:

Yes. but it is more attributed to the fat rich products.

The sentence has been corrected (line 47-51)

its an abbreviation, it should be revealed what does it abbreviate.

The authors agree with the reviewer´s comment. OMS has been described as “World Health Organization”  (line 51-52 )

Citation?

Line 88. Burns, D.A., Ciurczak, E.W. Handbook of Near-Infrared Analysis, 3rd ed.; Dekker Inc.: New York, NY, USA, 2008. (line 90)

Citation?

Citation has been added: Shepherd, K.D., Walsh, M.G. "Infrared Spectroscopy—Enabling an Evidence-Based Diagnostic Surveillance Approach to Agricultural and Environmental Management in Developing Countries," J. Near Infrared Spectrosc. 15, 1-19 (2007) (line 99)

sensory quality and maturity of peas

 That is done (line 105)

this paragraph does not fit into the intro

 The paragrah has been fitted at end of the Introduction section. (line 130-132)

I think it would be better to denote the pictures with a, b, c, d.... also in the caption.

 That is done (Figure 1) (line 140)

the abbreviation of these should be explained before this not after.

That is done (line 152-153)

From 166-169. its not clear how does is results in 144 samples.

The sentence has been corrected in the manuscript

For each cultivar (8), treatment (2) and replicate (3), three samples were used. Each sample was comprised of 5 pods from 3 plants selected at random, which then were averaged (line 172-174)

Therefore 144 samples were used: 8x2x3x3 =144

Did the authors test the normality? ANOVA can be only computed if the normality is good. Moreover, for choosing the optimal post-hoc test honogeneity of the variances (Levene-test) must be computed. Tukey test can be used only if it correct (based on mean >0.05). It can be easily done using SPSS. If homogeneity is not assumed, Games-Howell test could be used! Moreover, authors should considere using MANOVA insteas of ANOVA as we speak about composition. For what group variable was the ANOVA performed, please state it

The authors has included in the manuscript the following paragraph:

Analysis of variance ANOVA was used to compare differences among treatments for total marketable yield. Previously, normality and homocedasticity were tested using the Shapiro-Wilk and Levenne tests, respectively. For these analyses, Fisher´s least significant difference (LSD) test was used to make comparisons of treatments of most interest, using the 5% level of significance (line 252-258)

Ok, but what number did the authors choose? It could not be random, as the spectra of the same repetition of the sample must stay in one batch...

The following paragraph has been added to clarify the sentence:

This number was proposed by WinISI software (Infrasoft International, Port Matilda, PA, USA), being five groups used as the calibration set and then tested on the remaining of the samples performing an validation. This process continued until each group of the six was used as validation group. WinISI software uses principal component analysis as a tool for selecting samples (spectra) to establish the calibration and validation groups. Thus, both groups are composed with samples representative of the whole spectral variability of the population with similar mean and standard deviations for each trait. (line 291-298)

This figure is not good! The standard deviaton must be on the figure!!!!!!!!!

 That is done. Figure 3 has been renamed Figure 4 in the revised version.  (line 375)

it is with capital here, but with lowercase in the table?

That is done (table 2, 3 and 4).

The same a, b, c, d needed for the figures and into the caption!

 That is done. Figure 6 has been renamed Figure 7 in the revised version (line 601).

Reviewer 2 Report

This is an interesting research dealing with the use of VIS/NIR spectroscopy in determination of quality parameters (TPC, pH, protein content C, h, TSS, etc.) of mangetout. However revisions should be made in order to meet quality requirements. English language should be thoroughly revised, since I have experienced some troubles during reading this manuscript. Other comments are inserted directly into the pdf version.

Author Response

REVIEWER 2

This is an interesting research dealing with the use of VIS/NIR spectroscopy in determination of quality parameters (TPC, pH, protein content C, h, TSS, etc.) of mangetout. However revisions should be made in order to meet quality requirements. English language should be thoroughly revised, since I have experienced some troubles during reading this manuscript.

We really appreciate the comments and suggestion of the reviewer providing valuable feedback on our manuscript.

For the provided specific points we have made the requested modifications and hope the revised version would meet the requirements.

Other comments are inserted directly into the pdf version.

We provide a point-by-point response to the reviewer’s comments of the pdf file.

The requirements of the reviewer expressed in the pdf file throughout the article have been answered; some of them are detailed below:

why italic ?

We understand that a common name such as  “mangetout” should be unitalicised. Therefore, “mangetout” has been replaced by “mangetout” throughout the manuscript, also we have added the scientific name in the title.

requires the analysis.…

That is done (line 28)

please explain this abbreviation

            The abbreviation OMS has been replaced by World Health Organization (WHO) (line 51)

add more references and examples.

Two new references and other quality compounds have been added (total soluble solids, fiber, phytoprostanes and phytofurans) in the manuscript. (line 77)

García‐García, M. C.; del Río Celestino, M.; Gil‐Izquierdo, Á.; Egea‐Gilabert, C.; Galano, J. M.; Durand, T.; Oger, C.; Fernández, J.; Ferreres, F.; Domínguez‐Perles, R. The value of legume foods as a dietary source of phytoprostanes and phytofurans is dependent on species, variety, and growing conditions. Eur J Lipid Sci Tech 2019, 121, 8, 1800484.

El-Seifi, S. K., Hassan, M. A., El-Bassiouny, R. E. I., Elwan, M. W. M., & Nasef, I. N. (2014). Effect of maturity stage on physical and chemical characteristics and determination of harvest time of sugar snap pea pods. Journal of Plant Production5(2), 305-314.

such as...? Please mention constituents and parameters for the sake of clarity.

The authors have added the following sentence:

NIR spectroscopy in combination with chemometric analyses can be used for analysis of numerous components (protein, carbohidrates, carotenoid, minerals, glucosinolates, phenolics) and parameters (firmness, °Brix, acidity, colour) of the sample to be analysed (18-23). (line 92-95)

Although this paragraph is true, it should be clearly written and supported with references.

The following reference has been added (line 99)

Ozaki Y. Near-infrared spectroscopy—its versatility in analytical chemistry. AnalyticalSciences: the International Journal of the Japan Society for Analytical Chemistry.2012; 28:545–563.

This last paragraph should be clearly written.

That was done (line 130-132)

add Latin name

Pisum sativum L. ssp. arvense has been added in the title

some reference with the similar example ?

The following reference has been added: García et al., 2021 (line 155)

how many recordings per each sample were performed ?

Six replicate spectra were recorded for each sample and the average of the spectra was calculated. (line 260)

what was the drying conditions ?. Briefly describe. What about ground conditions ?? Desscribe.

The authors have included the information about the dried samples in the revised manuscript (lines 261-266).

The samples were lyophilized using freeze-drying equipment (Telstar LyoQuest, Germany), then ground in a mill (Janke & Kunkel, model A10, IKA®-Labortechnik) for about 20 s to pass through a 0.5mm screen, and stored at −80 ∘C until analysis.   The samples were freeze-dried to eliminate the strong absorbance of water in the infrared spectral region, which overlaps with important bands of nutritional compounds that are present in low concentration.

What was the type of NIR instrument, software, manufacturer, city, country

An spectrometer (Model 6500 Foss-NIRSystems, Inc., Silver Spring, MD, USA) was used for registrating the spectra (line 268)

add reference

Shenk J.S.; Westerhaus, M.O. Population structuring of near infrared spectra and modified partial least squares regression. Crop Sci. 1991, 31, 1548-1555 (line 273)

For the sake of clarity what are dependent (X) and what are independent (Y) variables ? Please add.

The reference values were plotted as the dependent variable and the predicted NIRS values plotted as the independent variable. (line 276-278)

subchapters 2.5. and 2.6. should be concisely and clearly written. For instance: what was the first step and why.

What was the second step ? One sentence describing cross-validation method (plus reference one or two) and what is the purpose of this method. Add a number of samples used for cross-validation. Then which parameters are calculated in order to assess performances of the calibration equations obtained from cross-validation. The next step is to mention what the values of the calculated parameters mean considering obtained calibration equations.

The third step is external validation. Definition, number of samples used for external validation, which parameters are computed in order to demonstrate how well the calibration model predicts data. Ranges of R2 values for the external validation are important. Ranges of RPD are important as well (unsuitable, suitable, etc,).

The subchapters 2.5 and 2.6 have been rewritten according the reviewer´s comment.

these symbols M, L, m do not apply in the equation describing coefficient of determination in the cross-validation

The authors agree with the reviewer and the description has been added in the manuscript. (line 348-349)

yˆ=NIR measured value; y=mean ‘y’ value for all samples; yi=lab reference value for the ith sample.

kg (not Kg). Correct this throughout the whole manuscript.

That is done (line 375)

add values

The values have been added (0.55-0.65 kg m-2) (line 380)

please extend this paragraph with some additional information. What dou you mean by the terms enhancing crop quality and economizing water ? What comprises organic cultural ??? practices ?

A paragraph  with additional  information has been added: “In this regard, previous studies have showed that yield and quality of snap pods can be significantly affected by different compositions of fertilization [50, 51] and by different doses of water in the fertigation solution [13, 49].” (line 387-388)

The European regulation containing the authorized practices has been referenced in the paragraph. (line 383)

Why did you put calibration and cross-validation statistics together with the physo-chemical parameters ?

The mean and standard deviation for different parameters have been included in Table 2 and separated of calibration and cross validation statistics (Table 3).

seven cultivars ? I don't understand

We agree with the comment and we have corrected the sentence

Also add the meanings for the abbreviations related to physico-chemical parameters under the table.

The meanings for the abbreviations related to physic-chemical parameters have been included under the table 3 in the revised version.

I would suggest to delete measured by FNS-6500. If I figured correctly, FNS-6500 is an NIR instrument.

That was done

what is crocanti pod ?

“Crocanti” has been replaced by “crunchy” (line 414)

define complete sample set.

The sentence was unclear and it has been improved: A remarkable variability in the VIS region (400–850 nm) absorbance spectra was observed because to pigments. (line 472-474)

unclear sentence

The sentece has been rewritten and we have included a reference.

The peak around 640-700 nm illustrated the colour transition of pea pods correlated with the chlorophyll content that absorbs radiation in this region (Merzlyak et al., 2003). (line 475-477)

Merzlyak, M.N., Solovchenko, A.E. & Gitelson, A.A. (2003). Reflectance spectral features and non-destructive estimation of chlorophyll, carotenoid and anthocyanin content in apple fruit. Postharvest Biology and Technology, 27(2), 197-211.

be more specific; what is the reason for the significant abosrption bands in the wavelength range 1300-2000, 2200 - 2400, 1400-2500 nm ?

Extend this paragraph and support with reference, if necessary.

The paragraph has been improved and supported with reference:

Absorption bands in the region from 1300 to 2000 nm have been assigned to 3rd overtones of C-N (amines); C=O (ketones, amino acids); C-O (long-chain fatty acids, phenols). From 2200 to 2400 nm, absorptions of C-N (primary amines); C-O (alcohols) have been assigned to the 3rd overtones of these compounds, while in the same region, C-H (asymmetrical deformation) and C-O (symmetrical vibrations) have been assigned to the 2nd overtones of these molecules. Finally, 2nd overtones of C-H deformation and C-N (amides) have been reported in the 2400-2500 nm region (Murray and Williams. Chemical principles of Near-infrared technology, In Near-Infrared technology in the agricultural and foods industries, 1987, pp-17-34. Edited by Phil Williams and Karl Norris. American Association of cereal Chemists, Inc. St. Paul Minnesota, USA). (line 478-484)

Legend is missing

That was done

It should be refreshed with some newer references.

 That was done

Reviewer 3 Report

The authors have presented a research work on the application of visible and near infrared spectroscopy in determining various quality parameters of mangetout. The number of samples used in the study and the methods in selecting the samples does carry a high technical merit especially in spectroscopy study of food/fruits products. However, in the current state, the manuscript contains several limitations that need to be addressed thoroughly before the manuscript can be reconsider for possible publication in Sensors.

  1. The authors need to include the real image of the reflectance spectroscopy measurement on the sample.
  2. Section 2.4 and 3.3 are named as NIRS analysis while the title of the manuscript suggest that the research should covers the entire VIS-NIR wavelengths. Does the calibration and validation analyses were derived only using NIRS wavelengths?
  3. Why the authors didn’t include lightness (L) parameter in their colorimetry analysis?
  4. Chroma and hue are the colorimetry representation of the samples color. But it is unclear why did the authors try to predict the color measurement using NIRS? The chroma and hue values may be included as the reference on the physical attributes of the mangetout samples. But to predict them by using any of the NIRS wavelengths are meaningless.
  5. The authors need to further elaborate on the methods (preferably with illustration/image) they employ in measuring TSS, pH and firmness of the samples since the size of the sample is small and lack of water content.
  6. How many samples per each cultivar were used in the study? Comparing quality parameters from each cultivar and fertigation treatment will raised the technical quality of the data presentation.
  7. For Figure 6, please include slope and bias for each of the graph.
  8. Please also provide graphs for calibration. It would be interesting to see the distribution of all 144 data (from Table 2 results).
  9. Please use larger font size to label your x and y-axis in Figure 6.
  10. The number of references presented in the manuscript are very significant. However, I would further suggest the authors to also include the results of NIRS application in measuring variety of legumes for comparative study.
  11. Please restructure your paragraphs in the Introduction section and data analysis. The paragraph should begin with a topics sentence and strengthen by several supporting details from various references. In the current state, there are many short paragraph where several of them are not in harmony with each other.

Author Response

REVIEWER 3

The authors have presented a research work on the application of visible and near infrared spectroscopy in determining various quality parameters of mangetout. The number of samples used in the study and the methods in selecting the samples does carry a high technical merit especially in spectroscopy study of food/fruits products. However, in the current state, the manuscript contains several limitations that need to be addressed thoroughly before the manuscript can be reconsider for possible publication in Sensors.

We really appreciate the comments and suggestion of the reviewer providing valuable feedback on our manuscript.

For the provided specific points we have made the requested modifications and hope the revised version would meet the requirements.

  1. The authors need to include the real image of the reflectance spectroscopy measurement on the sample.

 A real image of the reflectance spectroscopy measurement on the sample has been included in the Figure 3  (line 201)

  1. Section 2.4 and 3.3 are named as NIRS analysis while the title of the manuscript suggest that the research should covers the entire VIS-NIR wavelengths. Does the calibration and validation analyses were derived only using NIRS wavelengths?

 The authors thank the comments of the reviewer. The entire VIS-NIRS wavelengths were used for both calibration and validation analyses. Therefore, attending the consideration made by the reviewer, authors have changed the 2.4 and 3.3 sections as follow:  2.4. VIS-NIRS analysis (line 259);  3.3. VIS-NIRS analysis (line 470)

  1. Why the authors didn't include lightness (L) parameter in their colorimetry analysis?

The Ligtness data exhibited a narrow range which was not sufficient to develop calibration equation for this parameter.

  1. Chroma and hue are the colorimetry representation of the samples color. But it is unclear why did the authors try to predict the color measurement using NIRS? The chroma and hue values may be included as the reference on the physical attributes of the mangetout samples. But to predict them by using any of the NIRS wavelengths are meaningless.

 Based on our previous experience we believe that the inclusion of NIR wavelengths in addition to VIS has provided more information in the prediction of colour parameters than if we only included the VIS part. Because there are chemical compounds (e.g. cellulose) in the pea pod and seed that are related to the colour, and therefore correlate in a secondary way with the Hue and Chroma parameters. The color is caused by reflection of helicoidally stacked cellulose microfibrils that form multilayers in the cell walls of the epicarp (Vignolini et al 2012). Thus the wavelength at 1932 nm (O-H stretching plus O-H deformation) could be related to the cellulose of the pod tissues as can be observed in the MPLS loadings of the optimal calibrations for colour parameters (Figure 7).

Vignolini, S., Rudall, P.J., Rowland, A.V., Reed, A., Moyroud, E., Faden, R.B., Baumberg, J.J., Glover, B.J., Steiner U. Pointillist structural color in Pollia fruit. Proc Natl Acad Sci. 2012 ,109,15712-5.

  1. The authors need to further elaborate on the methods (preferably with illustration/image) they employ in measuring TSS, pH and firmness of the samples since the size of the sample is small and lack of water content.

 Following the reviewer´s comment an illustration has been included. (Figure 3 of the revised manuscript) (line 201)

  1. How many samples per each cultivar were used in the study? Comparing quality parameters from each cultivar and fertigation treatment will raised the technical quality of the data presentation.

 The sentence has been corrected in the manuscript

For each cultivar (8), treatment (2) and replicate (3), three samples were used. Each sample was comprised of 5 pods from 3 plants selected at random, which then were averaged (line 172-174)

Therefore 144 samples were used: 8x2x3x3 =144

The authors agree with the reviewer about that the technical quality of the data presentation would be raised comparing quality parameters from each cultivar and fertigation treatments. However, we intend to publish the data in an article we are writing as part of a doctoral thesis.

  1. For Figure 6, please include slope and bias for each of the graph.

           That is done in the Figure 7 of the revised manuscript (line 600)

8. Please also provide graphs for calibration. It would be interesting to see the distribution of all 144 data (from Table 2 results).

      That is done in the Figure 7 of the revised manuscript (line 600)

  1. Please use larger font size to label your x and y-axis in Figure 6.

      The Figure has been improved. Figure 6 has been renamed Figure 7 of the revised manuscript (line 600).

 10. The number of references presented in the manuscript are very significant. However, I would further suggest the authors to also include the results of NIRS application in measuring variety of legumes for comparative study.

The authors have included other NIRS applications in measuring variety of legumes (lines 582-585).

  1. Please restructure your paragraphs in the Introduction section and data analysis. The paragraph should begin with a topics sentence and strengthen by several supporting details from various references. In the current state, there are many short paragraph where several of them are not in harmony with each other.

Following the reviewer´s recommendations we have improved the Introduction section

Round 2

Reviewer 3 Report

Thanks to the authors who have taken a good initiative to improve their manuscript based on the comments given to them. I would like to suggest the authors to include an additional paragraph with supporting literature in their discussion to explain in details the contribution of NIR in color-pigments analysis. 

Author Response

Thanks to the authors who have taken a good initiative to improve their manuscript based on the comments given to them. I would like to suggest the authors to include an additional paragraph with supporting literature in their discussion to explain in details the contribution of NIR in color-pigments analysis.

Thank you very much for your positive comments on our manuscript and your suggestions. We have made the corresponding modifications according to your comments. The contribution of NIR in color analysis has been added in 3.3.4. section (lines 595-606):

“With respect to NIR region, previous studies have shown the contribution of this region to predict colour parameters for species such as fresh Ginkgo biloba leaves [88], green-leafy species [89] or Sassafras tzumu [90]. Some plant chemical compounds (e.g. phenolics and flavonoids) respond to the stress and environment changes and correlate in a secondary way with the colour parameters. The characteristic bands of phenolics and flavonoids can be detected in wavelength regions from 1415 nm to 1512 nm, 1650 to 1750 nm, and from 1955 to 2035 nm in the MPLS loadings for colour parameters (Figure 8) [91]. Futhermore, the colour is caused by reflection of helicoidally stacked cellulose microfibrils that form multilayers in the cell walls of the epicarp [92]. Thus, the wavelengths at 1932 nm (O-H stretching plus O-H deformation) could be related to the cellulose of the pod tissues as can be observed in the MPLS loadings of the optimal calibrations for colour parameters (Figure 8).”

The following references have also been added:

  1. Shi, J.Y; Zou, X.B; Zhao, J.W.; Mel, H.; Wang, K.L.; Wang, X.; Chen, H. Determination of total flavonoids content in fresh Ginkgo biloba leaf with different colors using near infrared spectroscopy. Spectrochimica Acta Part A: Molecular and Biomolecular Spectroscopy 2012 94, 271-276.
  2. Xue, L.; Yang, L. Deriving leaf chlorophyll content of green-leafy vegetables from hyperspectral reflectance. ISPRS J Photogramm Remote Sens 2009 64(1), 97–106.
  3. Li, Y.; Sun, Y.; Jiang, J.; Liu, J. Spectroscopic determination of leaf chlorophyll content and color for genetic selection on Sassafras tzumu. Plant Methods 2019 15. 10.1186/s13007-019-0458-0.
  4. Verardo, V.; Cevoli, C.; Pasini, F.; Gómez-Caravaca, A. M.; Marconi, E.; Fabbri, A.; Caboni, M. F. Analysis of oligomer proanthocyanidins in different barley genotypes using high-performance liquid chromatography-fluorescence detection-mass spectrometry and near-infrared methodologies. J Agric Food Chem 2015 63(16), 4130–4137.
  5. Vignolini, S.; Rudall, P.J.; Rowland, A.V.; Reed, A.; Moyroud, E.; Faden, R.B.; Baumberg, J.J.; Glover, B.J.; Steiner U. Pointillist structural color in Pollia fruit. Proc Natl Acad Sci. 2012, 109, 15712-5.
